# Serum KL-6 Could Represent a Reliable Indicator of Unfavourable Outcome in Patients with COVID-19 Pneumonia

**DOI:** 10.3390/ijerph18042078

**Published:** 2021-02-20

**Authors:** Riccardo Scotto, Biagio Pinchera, Francesco Perna, Lidia Atripaldi, Agnese Giaccone, Davide Sequino, Emanuela Zappulo, Alessia Sardanelli, Nicola Schiano Moriello, Anna Stanziola, Marialuisa Bocchino, Ivan Gentile, Alessandro Sanduzzi

**Affiliations:** 1Section of Infectious Diseases, Department of Clinical Medicine and Surgery, University of Naples Federico II, 80131 Naples, Italy; ri.scotto@gmail.com (R.S.); agnesegiaccone94@gmail.com (A.G.); e.zappulo@gmail.com (E.Z.); ale_sardanelli@hotmail.com (A.S.); veghan@gmail.com (N.S.M.); ivan.gentile@unina.it (I.G.); 2Section of Respiratory Diseases, Department of Clinical Medicine and Surgery, Monaldi Hospital, University of Naples Federico II, 80131 Naples, Italy; federico90@libero.it (F.P.); davesequino@gmail.com (D.S.); annaagnese.stanziola@unina.it (A.S.); marialuisa.bocchino@unina.it (M.B.); sanduzzi@unina.it (A.S.); 3Laboratory of Clinical Biochemistry, Monaldi Hospital, 80131 Naples, Italy; lidia.atripaldi@virgilio.it; 4Staff of United Nations Educational, Scientific and Cultural Organization (UNESCO), Health Education and Sustainable Development, University Federico II of Naples, 80131 Naples, Italy

**Keywords:** COVID-19, SARS-CoV-2, KL-6, pneumonia, mortality

## Abstract

KL-6 is a sialoglycoprotein antigen which proved elevated in the serum of patients with different interstitial lung diseases, especially in those with a poorer outcome. Given that interstitial pneumonia is the most common presentation of SARS-CoV2 infection, we evaluated the prognostic role of KL-6 in patients with COVID-19 pneumonia. Patients with COVID-19 pneumonia were prospectively enrolled. Blood samples were collected at the time of enrolment (TOE) and on day 7 (T1). Serum KL-6 concentrations were measured by chemiluminescence enzyme immunoassay using a KL-6 antibody kit (LUMIPULSE G1200, Fujirebio) and the cut-off value was set at >1000 U/mL. Fifteen out of 34 enrolled patients (44.1%) died. Patients with unfavourable outcome showed significantly lower P/F ratio and higher IL-6 values and plasmatic concentrations of KL-6 at TOE compared with those who survived (median KL-6: 1188 U/mL vs. 260 U/mL, *p* < 0.001). KL-6 > 1000 U/mL resulted independently associated with death (aOR: 11.29, *p* < 0.05) with a positive predictive value of 83.3%. Our results suggest that KL-6 is a reliable indicator of pulmonary function and unfavourable outcome in patients with COVID-19 pneumonia. A KL-6 value > 1000 U/mL resulted independently associated with death and showed good accuracy in predicting a poorer outcome. KL-6 may thus represent a quick, inexpensive, and sensitive parameter to stratify the risk of severe respiratory failure and death.

## 1. Introduction

Since its beginning, the pandemic outbreak of Coronavirus disease 2019 (COVID-19) involved 188 countries all over the world, exceeding 20 million cases and 745,000 deaths globally [1]. Facing this unprecedented emergency, the optimization of the healthcare resources and the identification of reliable predictors of clinical deterioration became of crucial importance.

Although SARS-CoV2 infection is associated with different clinical patterns, the lung proved to be specifically targeted and interstitial pneumonia represents the most frequent clinical feature of pulmonary involvement [2]. As regards therapy, great progress has been made in recent months, with the use of Dexamethasone, heparin and Remdesivir. Although several trials are still underway to evaluate new therapeutic approaches [3]. Furthermore, a very important resource is the recent introduction in the fight against SARS-CoV-2 infection of vaccination which is a fundamental step in the fight against the disease [4]. Several analytes, such as C-reactive protein (CRP) and IL-6, have been used in clinical practice to evaluate the inflammatory response and predict the clinical evolution of patients with COVID-19 pneumonia. Among the others, we focused on the significance of elevated Krebs von den Lungen 6 (KL-6) factor as a possible prognostic tool for disease activity and poorer outcome in these patients.

Serum levels of KL-6 are elevated in a variety of respiratory and non-respiratory conditions, including breast and pancreatic cancer [5,6] and diabetes mellitus [7]. However, most attention deserves its role in different interstitial lung diseases (ILDs), such as interstitial pneumonia [8], alveolar proteinosis [9], pulmonary sarcoidosis [10] and radiation pneumonitis [11]. In fact, KL-6 levels in the serum of patients with interstitial pneumonia are significantly higher compared to those of healthy people and patients with other respiratory diseases [12,13].

Immunohistochemical studies showed that KL-6 is a sialoglycoprotein antigen with a molecular weight equal to or greater than 1000 kDa strongly expressed on type II pneumocytes (reported in 1985 by Kohno et al.), and serum KL-6 levels are regarded as an index of alveolar epithelial cell damage and subsequent regeneration [14,15]. Moreover, the expression of KL-6 protein seems to correlate with altered alveolar-capillary permeability [16], suggesting a link between high KL-6 serum levels and alveolar epithelial barrier dysfunction, and the subsequent onset of acute respiratory distress syndrome (ARDS). A previous study examined KL-6/MUC1 levels in the serum and pulmonary epithelial lining fluid (ELF) or bronchoalveolar lavage fluid (BALF) of patients with ARDS or acute lung injury (ALI) [17,18]. These studies reported that the KL-6/MUC1 levels in these samples were significantly higher in non-survivors than in survivors. A recent study from our laboratory evaluated the levels of KL-6/MUC1 in ELF and serum obtained at multiple time points from patients with ARDS [15]. A comparison of the kinetics of KL-6/MUC1 levels in ELF and serum between survivors and non-survivors revealed that only the KL-6/MUC1 levels in ELF on days 0–3 after the diagnosis of ARDS were significantly higher in non-survivors than in survivors.

Furthermore, elevated KL-6 levels proved to be an independent predictor of acute exacerbation correlated to diffuse alveolar damage and significantly higher mortality rates in several ILDs such as idiopathic interstitial pneumonia, connective tissue diseases and drug-related interstitial fibrosis among the others [19,20,21,22,23]. The cut-off level usually adopted to define a baseline increase of serum KL-6 ranged between 700 and 1000 U/mL. Thus, we assumed that KL-6 levels might be a ready and reliable indicator of disease activity in COVID-19 patients, representing a useful tool to stratify the risk of severe pneumonia and ARDS in each case and establish the correct time for starting intensive care.

## 2. Materials and Methods

### 2.1. Study Design

Consecutive inpatients with a diagnosis of COVID-19 (SARS-CoV-2 infection) pneumonia admitted at University Hospital Federico II—Department of Clinical Medicine and Surgery, Section of Infectious Diseases, were enrolled. Patients with no evidence of interstitial pneumonia at chest X-ray (XR) or computer tomography (CT) exams were excluded, as well as those who denied informed written consent. Patients with a positive rhinopharyngeal swab for SARS-COV-2 RNA were diagnosed with COVID-19. Patients with COVID-19 and evidence of interstitial pneumonia at XR/CT were diagnosed with COVID-19 pneumonia. Blood samples from all the enrolled patients were collected at the time of enrolment (TOE) and on day 7 (T1) from TOE. A further blood sample from patients with no evidence of clinical improvement at day 14 from TOE was collected (T2). The following analytes were dosed on blood samples: KL-6, C-reactive protein (CRP), the white blood cells count (WBC), d-dimer, fibrinogen and IL-6. All patients underwent arterial blood gas test (ABG) at EOT, T1 and T2 (if performed).

### 2.2. KL-6 Dosage

Peripheral blood samples were centrifuged at 500× *g* for 10 min. The serum KL-6 level was measured by chemiluminescence enzyme immunoassay (CLEIA) using a KL-6 antibody kit (LUMIPULSE G1200, Fujirebio) according to the manufacturer’s protocol.

In brief, KL-6 present in the sample binds to microparticles coated with anti-KL-6 MoAb to form immunocomplexes. After the first washing aimed at discarding not bound materials, a second anti-KL-6 MoAb marked with alkaline phosphatase specifically binds to KL-6 present on particles.

After a second wash, the sample reacts with a substrate solution which utilizes the chemiluminescence phenomenon of 3-(2′-spiroadamantyl)-4-methoxy-4-(3″-phosphoryloxy)-phenyl-1,2-dioxetane (AMPPD). Luminescence is generated by the scission of dephosphorylated AMPPD and its intensity is proportional to KL-6 quantity.

Regarding the normal values, the reference intervals of serum KL-6 levels in the healthy control group is 273-458 U/mL (median + 1.96 SD). In this study, we adopted a cut-off value of 1000 U/mL as the most suitable to distinguish between active and stable disease and to predict survival [18,24,25].

### 2.3. Statistical Analysis

Data were reported as the median and interquartile range (IQR) given their non-parametric distribution. For correlation analysis, Pearson correlation was executed, while linear regression was performed for regression analysis. For both correlation and regression analysis, KL-6 values were chosen as the dependent variable. Correlation and regression analyses were performed on all laboratory results, regardless of the observation time, thus multiple values for each patient were included. For comparisons between continuous variables, the Mann-Whitney U test was performed. Logistic regression analysis was performed to evaluate the presence of risk factors for mortality. Co-variates significantly associated with death at the univariate analysis were also analysed in a multivariate model. The *p*-value for statistical significance was set at <0.05 for all the tests. The ROC curve was calculated setting the cut-off at >1000 U/mL. IBM SPSS© ver. 25.0 was used for statistical analysis.

### 2.4. Ethical Statement

With respect to the ethical issues, the study was conducted in compliance with the Declaration of Helsinki and the principles of good clinical practice. The authors confirm that the ethical policies of the journal, as noted on the journal’s author guidelines page, have been adhered to. The study was evaluated and approved by the Ethics Committee of the National Relief Hospital (A.O.R.N.)—Ospedali dei Colli of Naples. The study was approved on 26 June 2020 with number 524. We underline that KL-6 was measured only on samples taken routinely for clinical reasons and with the patient’s consent

## 3. Results

Thirty-four patients with a diagnosis of COVID-19 pneumonia were enrolled. Clinical characteristics of the enrolled patients are reported in Table 1. Most patients were male (67.6%) while the median age was 63 years (IQR: 54–71). Thirty-two out of 34 patients (94.1%) were receiving oxygen therapy at TOE, with a median PaO2/FiO2 ratio of 236 (IQR: 184–335). Nine out of 34 patients (26.5%) had a P/F ratio of <200 at TOE. Median KL-6 value at TOE was 411 U/mL (IQR: 177–1192). Table 2 and Table 3 show the data of the patients at 7 days and 14 days from the time of enrolment, respectively.

At correlation and regression analysis, KL-6 values were found to be inversely correlated with P/F ratio (ρ = −0.337, *p* < 0.05, r^2^ = 0.113), while they were significantly correlated with IL-6 values (ρ = 0.402, *p* < 0.01, r^2^ = 0.162). KL-6 was also found to be correlated with WBC and lymphocyte counts (Table 4).

Among the 34 patients enrolled, 15 (44.1%) died. Patients who had an unfavourable outcome showed higher plasmatic concentrations of KL-6 at TOE compared with those who survived (median KL-6: 1188 U/mL [IQR: 592–3680] vs. 260 U/mL [IQR: 125–421], *p* < 0.001). Patients with unfavourable outcome also showed lower P/F ratio compared with patients who survived (median P/F ratio: 198 [IQR: 168–239] vs. 300 [IQR: 234–350], *p* < 0.01) as well as higher IL-6 values (median IL-6: 810 pg/mL [IQR: 51.5–1420.0] vs. 16.5 pg/mL [IQR: 8.4–62.8], *p* < 0.01). There were no differences in the distribution of other laboratory parameters at TOE (Table 5).

At the logistic regression analysis for mortality, a KL-6 value above 1000 U/mL showed to be significantly associated with death (OR: 17.00, 95CI: 2.76–104.54, *p* < 0.01) together with a P/F ratio < 200 and IL-6 > 100 pg/mL (Table 6). At the multivariate analysis, only KL-6 > 1000 U/mL was independently associated with death (aOR: 11.29; 95CI: 1.04–122.00, *p* < 0.05). The overall accuracy of KL-6 values at TOE to predict an unfavourable outcome was good (AUC: 0.849, 95CI: 0.702–0.996, *p* < 0.01, Figure 1). When the cut-off for positivity was set at >1000 U/mL, KL-6 showed a positive predictive value for the death of 83.3% (95CI: 70.8–95.8%) and a negative predictive value of 77.3% (95CI: 63.3–91.3%).

## 4. Discussion

Since the first outbreak of COVID-19 in Italy, it was clear that the number of patients requiring intensive treatment outnumbered the availability of ICU beds in many areas of the country. Thus, the identification of reliable predictors of clinical deterioration is of major concern for the clinicians to stratify the risk for each patient and focus the treatment efforts on those with a worse expected clinical evolution. In this perspective, many clinical, radiologic and laboratory features have been taken into account and showed a significant correlation with a poorer outcome in patients with COVID-19 disease. A study conducted at Humanitas Research Hospital in Rozzano (Milan, Italy) evaluated several baseline parameters on a large retrospective cohort of patients diagnosed with COVID-19, pointing out that the ratio of PaO2 to FiO2 (P/F), the lymphocyte count and the serum levels of C-Reactive Protein (CRP) and Interleukin 6 (IL-6) were associated with the risk of clinical deterioration [26]. Similar findings regarding the serum levels of CRP and IL-6 have been reported by other retrospective studies, suggesting that the cytokine storm might play a major role in the pathogenesis of the lung disease progression in patients with SARS-CoV2 pneumonia, thus representing a useful tool to predict the clinical course of these patients [27,28].

According to former evidence that Krebs von den Lungen, 6 (KL-6) concentrations correlate both with the pulmonary function and the disease extent, we speculated that it might also represent a reliable predictor of severity in COVID-19 interstitial pneumonia [29]. KL-6 is a mucin-like, high molecular weight glycoprotein expressed on the surface membrane of alveolar epithelial cells (AEC-II) and bronchiolar epithelial cells. It represents a solubilized component in the pulmonary epithelial lining fluid mainly produced by damaged or regenerating alveolar type II pneumocytes. Consequently, KL-6 concentration proved elevated in the serum of patients with Interstitial Lung Disease (ILD) mainly owing to [13,24]:Elevated pulmonary production due to diffuse hyperplasia of alveolar epithelial cells, also called alveolar pneumocytes.Increase in spillover towards the systemic circulation due to leakage of the integrity of the alveolo-capillary membrane.

More than 350 papers investigating the clinical significance of KL-6 in Interstitial Lung Disease (ILDs) have been published thus far. These studies suggested that KL-6 serum levels are useful in the detection of AEC injury, as well as in the evaluation of disease activity and the prediction of clinical outcomes in different types of ILDs [30]. Previous studies indeed showed that initial serum KL-6 above 1000 U/mL and serial increases in serum KL-6 are associated with shorter survival; on the other hand, initial serum KL-6 lower than 1000 U/mL and no serial increases correlate with better outcome [31]. Besides, KL-6 may be helpful in other clinical conditions, such as the prediction of Acute Exacerbation (AE) [32] or screening and monitoring of ILD associated with Connective Tissue Disease (CTD-ILD) [33]. In a prospective study conducted on sixty patients with COVID-19 and hospitalized at Siena COVID Unit University Hospital, it was observed that KL-6, in addition to being a prognostic marker for identifying patients with severe disease requiring mechanical ventilation, was found to be a predictive marker of the possible development of pulmonary fibrosis in post-COVID-19 [34].

KL-6 serum levels in COVID-19 patients showed to be significantly higher in severe and critically severe disease compared to mild disease and showed an association with the extent of the pulmonary lesions at CT scan in previous studies [35,36]. In our cohort, we evaluated the KL-6 concentration in the serum of patients with SARS-CoV2 interstitial pneumonia at TOE (T0) and day 7 from TOE (T1). Regardless of the time of observation, KL-6 values were found to be inversely correlated with P/F ratio and significantly correlated with IL-6 values, WBC and lymphocyte count, suggesting that KL-6 is a reliable indicator of pulmonary function as well as inflammation degree, which are an expression of the progression of COVID-19 disease. The patients were thereafter allocated into two different groups according to the outcome. Patients with unfavourable outcome showed a significantly higher KL-6 concentration at TOE compared with those who survived, as well as higher IL-6 values and a lower P/F. Although all KL-6, P/F and IL-6 seemed to be highly predictive of a poor outcome, only elevated KL-6 serum concentration resulted independently associated with death in our multivariate analysis. Consequently, the positive predictive value of KL-6 proves reliable (83.3%; 95CI: 70.8–95.8%), since most patients who showed a remarkably high serum titre died within a few days.

Our study showed that KL-6 might represent an easily detectable and sensitive parameter to stratify the risk of severe respiratory failure in patients with SARS-CoV2 interstitial pneumonia being a reliable index of lung disease severity. A cut-off of KL-6 > 1000 U/mL showed to be adequate in the prediction of clinical deterioration and it may help to establish the correct timing for initiating intensive care. It has also been suggested that monitoring KL-6 values in cases of COVID-19 ARDS might help to distinguish between patients with preserved lung compliance (type L) and patients with low lung compliance (type H), since KL-6 correlates with type II pneumocytes impairment and loss of lung elastance, thus allowing the clinicians to predict the response to mechanical ventilation [37]. Given its formerly reported importance for diagnosing, monitoring and evaluating the prognosis of pulmonary fibrosis in ILDs, the results achieved hitherto cannot exclude a possible, adjunctive role of KL-6 in estimating the fibrotic evolution of COVID-19 interstitial pneumonia in patients who recovered from acute disease [12,13,20]. However, further prospective studies are needed to assess the reliability of this biomarker.

Our study presents several limitations. It is conducted on a small cohort of patients from a single centre and regardless of possible underlying comorbidities that might have impacted the outcome. Nonetheless, our findings support the evidence of a major role played by KL-6 factor in predicting survival of patients with SARS-CoV2 interstitial lung disease. Moreover, being a ready, sensitive and inexpensive marker, KL-6 could effectively integrate the clinical and radiologic evaluation of the patients at the baseline and drive the physicians’ decision-making process. Since our results support the evidence of an important role played by factor KL-6 in predicting survival of patients with interstitial lung disease and since KL-6 is and being a ready, sensitive and inexpensive marker, our goal would be to evaluate the use of KL-6 to complement the clinical and radiological assessment of patients at baseline and to explore the hypothesis that this marker can guide the decision-making process of physicians. To achieve this aim, our future commitment will be to continue our study with the expansion of the sample. The main limitations of our study are two: first of all the number of subjects enrolled, and furthermore, it would have been more interesting the evaluation of the KL-6 trend at shorter time intervals.

## Figures and Tables

**Figure 1 ijerph-18-02078-f001:**
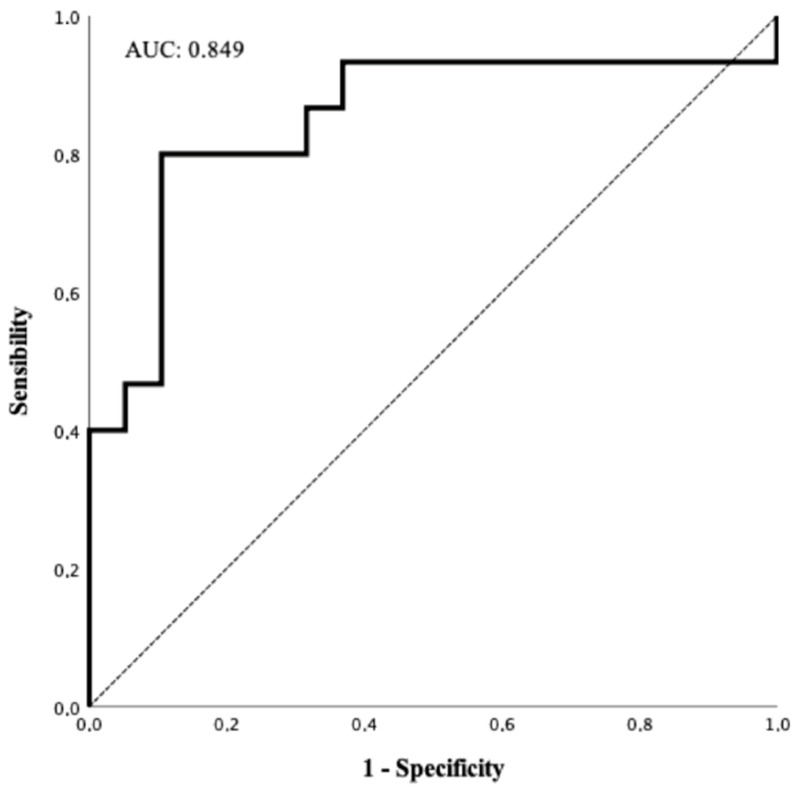
ROC curve for KL-6 diagnostic accuracy for death among patients with COVID-19 (n = 34).

**Table 1 ijerph-18-02078-t001:** Clinical and laboratory characteristics of the enrolled patients at the time of enrolment (n = 34).

	TOE (n = 34)
**KL-6 (U/mL; median, IQR)**	411 (177–1192)
**P/F ratio (n, %)**	
**≥300**	14 (41.2)
**250–299**	3 (8.8)
**200–249**	8 (23.5)
**<200**	9 (26.5)
**WBC (cell/µL; median, IQR)**	7400 (4715–10,372)
**Neutrophil count (cell/µL; median, IQR)**	6090 (4270–8210)
**Lymphocyte count (cell/µL; median, IQR)**	1380 (870–1570)
**D-dimer (mg/L; median, IQR)**	1.36 (0.70–1.94)
**CRP (mg/dL; median, IQR)**	3.82 (1.55–6.96)
**IL-6 (pg/mL; median, IQR)**	24.2 (9.37–102.85)

KL-6: Krebs von den Lungen 6, U: Unit, IQR: InterQuartile Range, P/F: Pa02/FiO2, WBC: White Blood Cells, CRP: C-Reactive Protein, IL-6: Interleukin 6.

**Table 2 ijerph-18-02078-t002:** Clinical and laboratory characteristics of the enrolled patients at the time of 7 days from enrolment—T1 (n = 34).

	T1 (n = 34)
**KL-6 (U/mL; median, IQR)**	570 (70–7580)
**P/F ratio (n, %)**	
**≥300**	6 (17.6)
**250–299**	7 (20.5)
**200–249**	6 (17.6)
**<200**	15 (44.3)
**WBC (cell/µL; median, IQR)**	6600 (1560–9850)
**Neutrophil count (cell/µL; median, IQR)**	5430 (950–7680)
**Lymphocyte count (cell/µL; median, IQR)**	1130 (370–1680)
**D-dimer (mg/L; median, IQR)**	1 (0.3–35.3)
**CRP (mg/dL; median, IQR)**	3.24 (0–18.8)
**IL-6 (pg/mL; median, IQR)**	73.6 (2–2476)

KL-6: Krebs von den Lungen 6, U: Unit, IQR: InterQuartile Range, P/F: Pa02/FiO2, WBC: White Blood Cells, CRP: C-Reactive Protein, IL-6: Interleukin 6.

**Table 3 ijerph-18-02078-t003:** Clinical and laboratory characteristics of the enrolled patients at the time of 14 days from enrolment—T2 (n = 15).

	T2 (n = 15)
**KL-6 (U/mL; median, IQR)**	296 (137–5548)
**P/F ratio (n, %)**	
**≥300**	7 (53.3)
**250–299**	5 (33.3)
**200–249**	1 (6.7)
**<200**	1 (6.7)
**WBC (cell/µL; median, IQR)**	6550 (2510–9470)
**Neutrophil count (cell/µL; median, IQR)**	3960 (1500–7220)
**Lymphocyte count (cell/µL; median, IQR)**	1530 (670–2750)
**D-dimer (mg/L; median, IQR)**	0.38 (0–13.10)
**CRP (mg/dL; median, IQR)**	0.5 (0.2–1.2)
**IL-6 (pg/mL; median, IQR)**	11.6 (2–131)

KL-6: Krebs von den Lungen 6, U: Unit, IQR: InterQuartile Range, P/F: Pa02/FiO2, WBC: White Blood Cells, CRP: C-Reactive Protein, IL-6: Interleukin 6.

**Table 4 ijerph-18-02078-t004:** Correlation and regression analysis of laboratory parameters among the enrolled patients, regardless of the time of observation.

Covariates	ρ Coefficient	*p*-Value	r^2^
KL-6 (dependent)	-	-	-
**P/F ratio**	**−0.337**	**<0.05**	**0.113**
**WBC**	**0.358**	**<0.05**	**0.128**
Neutrophil count	0.275	0.11	0.076
**Lymphocyte count**	**0.457**	**<0.01**	**0.209**
D-dimer	−0.006	0.973	0.000
CRP	−0.280	0.874	0.001
**IL-6**	**0.402**	**<0.01**	**0.162**

KL-6: Krebs von den Lungen 6, U: Unit, IQR: InterQuartile Range, P/F: Pa02/FiO2, WBC: White Blood Cells, CRP: C-Reactive Protein, IL-6: Interleukin 6.

**Table 5 ijerph-18-02078-t005:** Laboratory parameters distribution at TOE among patients with favourable and unfavourable outcomes.

Laboratory Parameters at TOE	Favourable Outcome (n = 19)	Unfavourable Outcome (n = 15)	*p*-Value
**KL-6 (U/mL; median, IQR)**	**260 (125–421)**	**1188 (592–3608)**	**<0.001**
**P/F ratio (median, IQR)**	**300 (234–350)**	**198 (168–239)**	**<0.01**
WBC (cell/µL; median, IQR)	7350 (4160–8810)	11560 (6510–14480)	0.189
Neutrophil count (cell/µL; median, IQR)	5795 (3155–7625)	9160 (4970–12160)	0.233
Lymphocyte count (cell/µL; median, IQR)	1270 (815–1495)	1640 (1130–1810)	0.233
D-dimer (mg/L; median, IQR)	1.3 (0.8–1.6)	1.7 (0.3–2.0)	0.900
CRP (mg/dL; median, IQR)	2.94 (1.46–5.55)	6.66 (4.16–28.6)	0.146
**IL-6 (pg/mL; median, IQR)**	**16.5 (8.4–62.8)**	**810.0 (51.5–1420.0)**	**<0.01**

KL-6: Krebs von den Lungen 6, U: Unit, IQR: InterQuartile Range, P/F: Pa02/FiO2, WBC: White Blood Cells, CRP: C-Reactive Protein, IL-6: Interleukin 6.

**Table 6 ijerph-18-02078-t006:** Univariate and multivariate regression analysis for mortality.

	OR (95CI)	*p*-Value	aOR (95CI)	*p*-Value
Male sex	1.60 (0.37–7.02)	0.53	-	-
Age > 60 years	1.17 (0.28–4.83)	0.83	-	-
KL-6 > 1000 U/mL	**17.00 (2.76–104.54)**	**<0.01**	**11.29 (1.04–122.00)**	**<0.05**
P/F ratio < 200	**20.57 (2.16–196.10)**	**<0.01**	3.11 (0.07–128.56)	0.550
IL-6 > 100 pg/mL	**9.37 (1.30–67.64)**	**<0.01**	4.03 (0.39–41.78)	0.243

OR: Odds Ratio, aOR: adjusted Odds Ratio, KL-6: Krebs von den Lungen 6, P/F: Pa02/FiO2, IL-6: Interleukin 6.

## Data Availability

Not applicable.

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
