# Peer review of "Serum KL-6 Could Represent a Reliable Indicator of Unfavourable Outcome in Patients with COVID-19 Pneumonia"

_ijerph, 2021, doi:10.3390/ijerph18042078_

Round 1

Reviewer 1 Report

In this paper, the authors deal about “Serum KL-6 could represent a reliable indicator of unfavourable outcome in patients with COVID-19 pneumonia”

Comments and suggestions:

  1. The introduction is too short and the connection with the title of the paper and the purpose of the paper is not clearly highlighted. There are very few data about SARS-CoV-2 and COVID-19 signs, no mention about potential treatment. Line34-41, I suggest adding short notions regarding these issues. ( org/10.1016/j.fct.2020.111769, doi.org/10.1007/s40199-020-00371-8., doi: 10.3389/fphar.2020.572870)
  2. The manuscript is non-attractive by readers, no cartoon or flowchart. Therefore, a graphical abstract/flowchart of the study is recommended.
  3. What are the clinical pitfalls and limitations of this study?
  4. Overall, the Manuscript is incomplete, unfinished, unpolished, non-attractive. Do the authors not have conclusions and future perspectives for this topic and their study?

Consider revising accordingly!

Author Response

Reviewer 1:

1) The introduction is too short and the connection with the title of the paper and the purpose of the paper is not clearly highlighted. There are very few data about SARS-CoV-2 and COVID-19 signs, no mention about potential treatment. Line34-41, I suggest adding short notions regarding these issues. (org/10.1016/j.fct.2020.111769, doi.org/10.1007/s40199-020-00371-8., doi: 10.3389/fphar.2020.572870)

R: We agree and beefed-up the introduction with the following paragraph:

“As regards therapy, great progress has been made in recent months, with the use of Dexamethasone, heparin and Remdesivir, although several trials are still underway to evaluate new therapeutic approaches. (Chandan Sarkar, Milon Mondal, Muhammad Torequl Islam, Miquel Martorell, Anca Oana Docea, Alfred Maroyi, Javad Sharifi-Rad , Daniela Calina. Potential Therapeutic Options for COVID-19: Current Status, Challenges, and Future Perspectives. Review Front Pharmacol. 2020 Sep 15;11:572870.) Furthermore, a very important resource is the recent introduction in the fight against SARS-CoV-2 infection  of vaccination which is a fundamental step in the fight against the disease. (Daniela Calina, Thomas Hartung, Anca Oana Docea, Demetrios A. Spandidos, Alex M. Egorov, Michael I. Shtilman, Felix Carvalho, and Aristidis Tsatsakis. COVID-19 vaccines: ethical framework concerning human challenge studies. Daru. 2020 Dec; 28(2): 807–812)

2) What are the clinical pitfalls and limitations of this study?

R: we added the following paragraph on the main limitations at the end of study design section

“The main limitations of our study are two: first of all the number of subjects enrolled, and furthermore it would have been more interesting the evaluation of the KL-6 trend at shorter time intervals”.

3) Overall, the Manuscript is incomplete, unfinished, unpolished, non-attractive. Do the authors not have conclusions and future perspectives for this topic and their study?

We added the following paragraph at the end of discussion section.

“Since our results support the evidence of an important role played by factor KL-6 in predicting survival of patients with interstitial lung disease and since KL-6 is a ready, sensitive and inexpensive marker, our goal would be to evaluate the use of KL-6 to complement the clinical and radiological assessment of patients at baseline to explore the hypothesis that this marker can guide the decision-making process of physicians. In order to achieve this aim, our future commitment will be to continue our study with the expansion of the sample.

Reviewer 2 Report

The paper by Scotto et al. is interesting as it focused on a really hot topic: the research of a reliable biomarker for the risk stratification and prognostic estimation in COVID-19 patients. Despite a certain lack of originality (the role of KL-6 and IL-6 in this setting have been already published) the paper is well written and provided further evidence on the potential utility of these biomarkers on this topic. 

I have only some minor comments:

  • you should show also the serum concentrations of KL-6, IL-6 and P/F at T1 and T2 and include them in the results analysis.
  • KL-6 in COVID-19 patients was associated not only to be related to disease severity and risk of mortality, but also as a predictor of lung fibrotic abnormalities at follow-up (https://pubmed.ncbi.nlm.nih.gov/33453011/). You should include and discuss this recent evidence in the Discussion

Author Response

Reviewer 2:

1) you should show also the serum concentrations of KL-6, IL-6 and P/F at T1 and T2 and include them in the results analysis.

We take the Reviewer’s point and, to address this issue, we included these results in table 2 and 3.

Table 2. Clinical and laboratory characteristics of the enrolled patients at time of 7 days from enrolment – T1 (n=34).

T1 (n=34)

KL-6 (U/mL; median, IQR)

570 (70-7580)

P/F ratio (n, %)

-         ≥ 300

-         250-299

-         200-249

-         < 200

6 (17.6)

                    7 (20.5)

6 (17.6)

15 (44.3)

WBC (cell/µL; median, IQR)

6600 (1560-9850)

Neutrophil count (cell/µL; median, IQR)

5430 (950-7680)

Lymphocyte count (cell/µL; median, IQR)

1130 (370-1680)

D-dimer (mg/L; median, IQR)

1 (0.3-35.3)

CRP (mg/dL; median, IQR)

3.24 (0-18.8)

IL-6 (pg/mL; median, IQR)

73.6 (2-2476)

Table 3. Clinical and laboratory characteristics of the enrolled patients at time of 14 days from enrolment – T2 (n=15).

T2 (n=15)

KL-6 (U/mL; median, IQR)

296 (137-5548)

P/F ratio (n, %)

-         ≥ 300

-         250-299

-         200-249

-         < 200

7 (53.3)

5 (33.3)

1 (6.7)

1 (6.7)

WBC (cell/µL; median, IQR)

6550 (2510-9470)

Neutrophil count (cell/µL; median, IQR)

3960 (1500-7220)

Lymphocyte count (cell/µL; median, IQR)

1530 (670-2750)

D-dimer (mg/L; median, IQR)

0.38 (0-13.10)

CRP (mg/dL; median, IQR)

0.5 (0.2-1.2)

IL-6 (pg/mL; median, IQR)

11.6 (2-131)

2) KL-6 in COVID-19 patients was associated not only to be related to disease severity and risk of mortality, but also as a predictor of lung fibrotic abnormalities at follow-up (https://pubmed.ncbi.nlm.nih.gov/33453011/). You should include and discuss this recent evidence in the Discussion

We agree and added the following paragraph in discussion section and quoted that paper.

In a prospective study conducted on sixty patients with COVID-19 and hospitalized at Siena COVID Unit University Hospital, it was observed that KL-6, in addition to being a prognostic marker for identifying patients with severe disease requiring mechanical ventilation, was found to be a predictive marker of possible development of plmonary fibrosis in post COVID-19. (Miriana d'Alessandro, Laura Bergantini, Paolo Cameli, Giuseppe Curatola, Lorenzo Remediani, David Bennett, Francesco Bianchi, Felice Perillo, Luca Volterrani, Maria Antonietta Mazzei, Elena Bargagli, Siena COVID. Serial KL-6 measurements in COVID-19 patients. Intern Emerg Med. 2021 Jan 16;1-5. doi: 10.1007/s11739-020-02614-7)

Furthermore, the correction of the name of Prof. Alessandro Sanduzzi is required. In particular, the name "Aeessandro" should read “Alessandro”.

We thanks all the reviewers for their suggestions-

Reviewer 3 Report

The authors evaluated the prognostic role of KL-6 in patients with COVID-19 pneumonia, and indicated that serum KL-6 was a sensitive parameter to stratify the risk of severe respiration failure and death for patients with COVID-19.

  1. As the authors mentioned in the introduction, serum levels of KL-6 are elevated in a variety of respiratory as well as non-respiratory conditions. However, there were not data regarding underlying disease such as cancer (breast and pancreatic) and diabetes in this study. The authors should show the data and compare unfavourable outcome with favourable outcome.
  2. In method, blood sample was collected at day 14 from TOE. But the authors did not show the data. Please show the results.
  3. Regardless the time of observation, KL-6 values were shown to be inversely correlated with P/F ratio and significantly correlated with some factors. Do the results mean that the patients included in this study did not cure during whole investigated duration?

Author Response

Reviewer 3:

1) As the authors mentioned in the introduction, serum levels of KL-6 are elevated in a variety of respiratory as well as non-respiratory conditions. However, there were not data regarding underlying disease such as cancer (breast and pancreatic) and diabetes in this study. The authors should show the data and compare unfavourable outcome with favourable outcome.

We agree and have summarized these data in discussion section.

2) In method, blood sample was collected at day 14 from TOE. But the authors did not show the data. Please show the results.

We agree that this was lacking. See also comment 1 to the reviewer 2

Table 2. Clinical and laboratory characteristics of the enrolled patients at time of 7 days from enrolment – T1 (n=34).

T1 (n=34)

KL-6 (U/mL; median, IQR)

570 (70-7580)

P/F ratio (n, %)

-         ≥ 300

-         250-299

-         200-249

-         < 200

6 (17.6)

                    7 (20.5)

6 (17.6)

15 (44.3)

WBC (cell/µL; median, IQR)

6600 (1560-9850)

Neutrophil count (cell/µL; median, IQR)

5430 (950-7680)

Lymphocyte count (cell/µL; median, IQR)

1130 (370-1680)

D-dimer (mg/L; median, IQR)

1 (0.3-35.3)

CRP (mg/dL; median, IQR)

3.24 (0-18.8)

IL-6 (pg/mL; median, IQR)

73.6 (2-2476)

Table 3. Clinical and laboratory characteristics of the enrolled patients at time of 14 days from enrolment – T2 (n=15).

T2 (n=15)

KL-6 (U/mL; median, IQR)

296 (137-5548)

P/F ratio (n, %)

-         ≥ 300

-         250-299

-         200-249

-         < 200

7 (53.3)

5 (33.3)

1 (6.7)

1 (6.7)

WBC (cell/µL; median, IQR)

6550 (2510-9470)

Neutrophil count (cell/µL; median, IQR)

3960 (1500-7220)

Lymphocyte count (cell/µL; median, IQR)

1530 (670-2750)

D-dimer (mg/L; median, IQR)

0.38 (0-13.10)

CRP (mg/dL; median, IQR)

0.5 (0.2-1.2)

IL-6 (pg/mL; median, IQR)

11.6 (2-131)

3) Regardless the time of observation, KL-6 values were shown to be inversely correlated with P/F ratio and significantly correlated with some factors. Do the results mean that the patients included in this study did not cure during whole investigated duration?

The results specified that 19 patients had a favorable outcome, while 15 had an unfavorable outcome during the study.

Furthermore, the correction of the name of Prof. Alessandro Sanduzzi is required. In particular, the name "Aeessandro" should read “Alessandro”.

We thanks all the reviewers for their suggestions-

Reviewer 3:

1) As the authors mentioned in the introduction, serum levels of KL-6 are elevated in a variety of respiratory as well as non-respiratory conditions. However, there were not data regarding underlying disease such as cancer (breast and pancreatic) and diabetes in this study. The authors should show the data and compare unfavourable outcome with favourable outcome.

We agree and have summarized these data in discussion section.

2) In method, blood sample was collected at day 14 from TOE. But the authors did not show the data. Please show the results.

We agree that this was lacking. See also comment 1 to the reviewer 2

Table 2. Clinical and laboratory characteristics of the enrolled patients at time of 7 days from enrolment – T1 (n=34).

T1 (n=34)

KL-6 (U/mL; median, IQR)

570 (70-7580)

P/F ratio (n, %)

-         ≥ 300

-         250-299

-         200-249

-         < 200

6 (17.6)

                    7 (20.5)

6 (17.6)

15 (44.3)

WBC (cell/µL; median, IQR)

6600 (1560-9850)

Neutrophil count (cell/µL; median, IQR)

5430 (950-7680)

Lymphocyte count (cell/µL; median, IQR)

1130 (370-1680)

D-dimer (mg/L; median, IQR)

1 (0.3-35.3)

CRP (mg/dL; median, IQR)

3.24 (0-18.8)

IL-6 (pg/mL; median, IQR)

73.6 (2-2476)

Table 3. Clinical and laboratory characteristics of the enrolled patients at time of 14 days from enrolment – T2 (n=15).

T2 (n=15)

KL-6 (U/mL; median, IQR)

296 (137-5548)

P/F ratio (n, %)

-         ≥ 300

-         250-299

-         200-249

-         < 200

7 (53.3)

5 (33.3)

1 (6.7)

1 (6.7)

WBC (cell/µL; median, IQR)

6550 (2510-9470)

Neutrophil count (cell/µL; median, IQR)

3960 (1500-7220)

Lymphocyte count (cell/µL; median, IQR)

1530 (670-2750)

D-dimer (mg/L; median, IQR)

0.38 (0-13.10)

CRP (mg/dL; median, IQR)

0.5 (0.2-1.2)

IL-6 (pg/mL; median, IQR)

11.6 (2-131)

3) Regardless the time of observation, KL-6 values were shown to be inversely correlated with P/F ratio and significantly correlated with some factors. Do the results mean that the patients included in this study did not cure during whole investigated duration?

The results specified that 19 patients had a favorable outcome, while 15 had an unfavorable outcome during the study.

Furthermore, the correction of the name of Prof. Alessandro Sanduzzi is required. In particular, the name "Aeessandro" should read “Alessandro”.

We thanks all the reviewers for their suggestions-

Round 2

Reviewer 1 Report

No answer given. 

Reviewer 3 Report

The authors revised appropriately. No further correction is necessary.